# AGPD: ADAPTIVE GUIDANCE POLICY DISTILLATION FOR IMITATION LEARNING

## ABSTRACT

Imitation Learning (IL) has been proven to be effective in training a policy network for long-horizon tasks under complex, unstructured environments. However, its effectiveness heavily relies on access to large-scale, high-quality datasets. To tackle this challenge, we introduce AGPD (**A**daptive **G**uidance **P**olicy **D**istillation for Imitation Learning), a novel policy distillation framework that transfers knowledge from a pretrained Diffusion Model (DM) to a student policy.

AGPD utilizes the DM to generate samples with diverse state observations for training the student policy. Unlike conventional distillation approaches, AGPD introduces a mechanism where the DM teacher generates samples by explicitly modeling the discrepancy between student and teacher policies. Furthermore, we theoretically prove that this generation approach explores a maximally large transition diversity with a smaller action divergence, thereby outperforming methods that generation through action mixing.

Moreover, AGPD incorporates a discriminator to assess policy quality and encourage students to mimic behaviors that closely align with the teacher, further enhancing learning from mixed-quality data.

Finally, experiments conducted in robotic simulations and real-world environments with multiple challenging tasks demonstrate the effectiveness of AGPD in distilling the teacher policy and self-distilling the student policy.

## 1 INTRODUCTION

Learning-based agent policies have shown great promise, achieving significant success in various applications O'Neill et al. (2023a); Liu et al. (2022); Ha et al. (2024). Imitation learning (IL) is a key method therein Chi et al. (2023). However, it still faces a critical challenge in real-world applications: **when demonstration data for specific tasks is scarce, the performance of imitation learning policies significantly degrades.** Mu et al. (2025) When only a limited amount of data is available, the policy is more likely to encounter out-of-distribution states during execution. As the agent diverges from the demonstration trajectory, this effect worsens over time, leading to cascading errors and a drop in performance—this is known as the Accumulation of Errors. To solve this, two primary approaches have been identified: (1) Acquiring additional task data, (2) Policy distillation. In this study, we propose combining these approaches to overcome imitation learning limitations by collecting data from the policy distillation process to incrementally augment the original task dataset, enabling effective learning even with limited task data.

Previous work, such as DAgger Ross et al. (2011), combines teacher and student policies to interact with the environment, collecting diverse states and actions that are then added to the training dataset. However, simple weight mixing results in the generation of data with low quality. This dependency limits the scalability and practical applicability of online data correction methods. In contrast, experience replay offers a simpler and more scalable approach. For example, Policy Distillation Rusu et al. (2015) introduced an experience replay method, where an RL-based policy is first trained, and successful interactions are stored in a buffer to train the student policy. While these methods have achieved strong performance, they face a key limitation: **the teacher policy's behavior tends to be deterministic, which reduces the diversity of data collected through experience replay**. As a result, the quality of the generated data may not be high.

Figure 1: `AGPD` Framework. Our method consists of three main stages: The diffusion-based policy is guided by the student policy to generate diverse high-quality trajectories. Then, these generated trajectories are used to expand the dataset, ensuring the method remains effective even when original task demonstrations are limited. Through a comparison of the generated results, we demonstrate that AGPD, compared to standard policy distillation (PD), produces a more diverse trajectory distribution. Finally, we train a classifier to distinguish between the teacher and student policies, incorporating an adversarial loss as an additional component in imitation learning to encourage the student to emulate the teacher's behavior.

To address above challenges, we introduce `AGPD` (**A**daptive **G**uidance Regularized policy distillation for **I**mitation **L**earning). We propose a two-phase training process, which includes the Adaptive Guided Policy Distillation phase and the Adversarial Imitation Learning phase. In the first phase, a student model and a pre-trained diffusion model teacher interact with the environment. Based on the VAE representation divergence between their decisions, the pre-trained model is guided to generate decisions close to the student model. Successful trajectories are stored in a data buffer, which serves as an additional high quality dataset for improving the training of the student policy. In the second stage, a discriminator is introduced to distinguish between decision features on teacher and student models. This encourages the student to behave more like the teacher during the training process, which stabilize the learning process of the policy. Additionally, based on prior theoretical analysis Belkhale et al. (2023), we prove that `AGPD` can generate greater state diversity with smaller action diversity during data generation, thereby enhance the quality of generated data compared with other method. Our contributions are summarized as follows:

- We introduce `AGPD`, a novel approach for distilling knowledge from a pretrained diffusion model into the student policy.
- We propose a unique two-phase training process that combines adversarial imitation learning with adaptive teacher policy distillation, improving the stability and efficiency of policy learning.
- We proof that, compared to action mixing methods, `AGPD` can generate higher-quality data by introducing perturbations via policy noise, resulting in larger transition diversity.
- Experimental results demonstrate that `AGPD` significantly enhances model performance on long-horizon and precise manipulation tasks in both simulated and real-world environments. For instance, achieve optimal performance in tasks such as precisely inserting a tube into a rack.

## 2 PRELIMINARY

**Imitation Learning:** IL is an efficient approach for training a policy model $\pi_\theta$ using expert demonstrations. During interaction with the environment, the expert provides the corresponding action $a_t$ for state $s_t$, storing the trajectory $\tau$ in the replay buffer $\mathcal{T}$. The training process for the policy model follows an offline supervised learning framework, formulated as:

$$\pi_\theta = \arg\min_\theta \sum_{\tau \in \mathcal{T}} \sum_{\{s_t, a_t\} \in \tau} \mathcal{L}_{IL}(\pi_\theta(s_t), a_t), \tag{1}$$

**Diffusion Models:** Diffusion modelHo et al. (2020) is a subset of Markovian generative models, via the denoising process, it is able to deduce the realistic data $x_0$ from the Gaussian distribution $p(x_T)$:

$$p_\theta(x_{t-1}|x_t) := \mathcal{N}(x_{t-1}; \mu_\theta(x_t, t), \sigma^2) \tag{2}$$

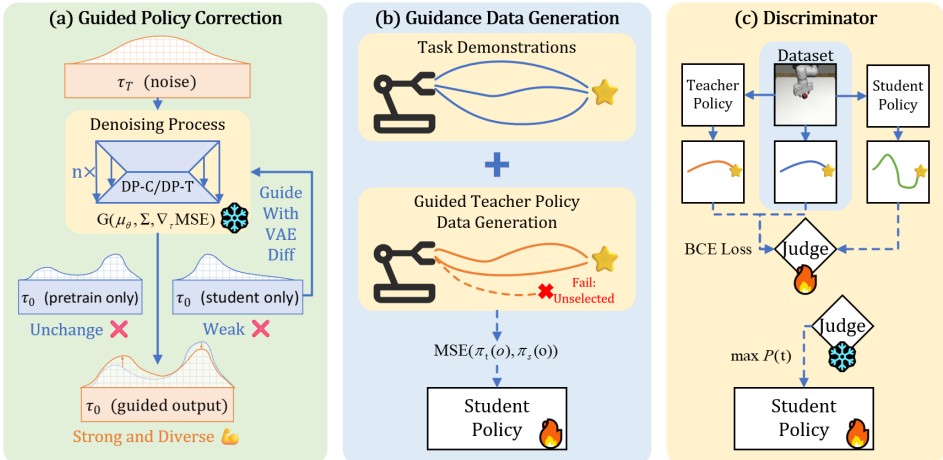

Figure 2: Details of `AGPD` . (a) Diffusion-based policy's output is guided by the student policy, ensuring diverse and reliable outputs. DP-C and DP-T represent CNN-based and transformer-based diffusion policies, respectively. (b) These diverse outputs are executed to augment the dataset, expanding the training data for the student policy. (c) Using recorded robot states from the dataset, a discriminator is trained to distinguish teacher and student policies, guiding the student to generate outputs closely aligned with the teacher's policy, thereby enhancing the student policy's performance.

where $\theta$, $\mu$ and $\sigma$ indicate the parameters of diffusion models, estimated mean, and constant variance. Given a measurement functionDhariwal & Nichol (2021), the direction of the gradients will guide the diffusion models to produce expected outputs.

$$p_\theta(x_{t-1}|x_t) := \mathcal{N}(x_{t-1}; \mu_\theta(x_t, t) + k\sigma^2 g, \sigma^2) \tag{3}$$

where $g = \nabla_{x_t} \log \phi(y|x_t)$ is the backward gradient.

**System Noise for State Diversity in Training Dataset:** Prior work Belkhale et al. (2023) proposed that the quality of imitation learning data can be evaluated through the diversity of state transitions, which is quantified by the *next-state coverage probability* $P_S(s; N, \epsilon)$ in the dataset. This metric characterizes the probability that, given an initial state $s$ and **system noise** $\sigma_s$ in the given environment , the next state generated by the learned policy falls within a specified distance of at least one next state sampled from the expert demonstration, where $N$ denotes the number of expert samples and $\epsilon$ represents the coverage tolerance threshold. Formally, this probability satisfies the inequality:

$$P_S(s; N, \epsilon) \geq 1 - \left[1 - \left(\frac{c\epsilon}{\sigma_s}\right)^d \exp(-a^2 d)\right]^N - \exp\left(-(a-1)^2 d\right), \tag{4}$$

where $c$ is the constant derived from the Gaussian distribution and $a$ is the policy action. According to the above equation, injecting the system noise will lead to higher coverage probability $P_S(s; N, \epsilon)$.

## 3 EFFICIENT NOISE INJECTION

As discussed in Section 2, injecting system noise into the environment enhances transition diversity. Similarly, policy noise (referred to as **Policy Noise**) introduces variability to expert actions during data collection, mitigating overfitting to narrow demonstrations.

Formally, policy noise is modeled as a Gaussian perturbation to expert actions: $a_N = a_E + \eta$, where $\eta \sim \mathcal{N}(0, \sigma_p^2 I)$. This perturbation increases per-state divergence but promotes transition diversity through dynamics, as analyzed below.

**Lemma 3.1.** [1] *Consider policy noise with variance $\sigma_p^2$. Assume state transitions follow $\rho(s'|s, a) = \mathcal{N}(\mu(s, a), \sigma_s^2 I)$. The effective variance of the transition distribution under noisy actions is $\sigma_{eff}^2 = \sigma_s^2 + \|\nabla_a \mu(s, a_E)\|^2 \sigma_p^2$.*

---

[1]Complete proofs for all theorems and lemmas are provided in the appendix.

**Theorem 3.2.** *Under the assumptions of Lemma 3.1, the probability of state coverage $P_S(s; N, \epsilon)$, as defined in Equation 4, satisfies:*

$$P_S(s; N, \epsilon) \geq 1 - \left[ 1 - \left( \frac{c\epsilon}{\sqrt{\sigma_{eff}^2}} \right)^d \exp(-a^2 d) \right]^N - \exp\left(-(a-1)^2 d\right). \quad (5)$$

In summary, system noise directly addresses coverage sparsity in finite datasets, while policy noise diversifies expert actions, indirectly enhancing transition diversity during data collection.

### 3.1 NOISE INJECTION IN DISTILLATION

Let $\pi_E(a|s)$ and $\pi_A(a|s)$ denote the expert and student policy, we consider the action of the student $\pi_A(a|s)$ as the noise in this paper, which is more likely to provide useful data for student training.

We introduce two effective noise injection methods for policy distillation: (1) *Action Mixing*, a linear interpolation between student and expert actions inspired by DAgger Ross et al. (2011), and (2) *Guidance Diffusion*, which employs gradient-based guidance to control the noise injection process. Our objective is to demonstrate that, for equivalent state diversity, guidance diffusion induces less action divergence than action mixing. These two methods can be defined as:

**Action Mixing:** The noised expert action is a convex combination between the student and expert:
$$a_{\text{mix}} = \lambda a_A + (1 - \lambda) a_E, \quad \lambda \in [0, 1], \quad (6)$$

**Guidance Diffusion:** The expert action is sampled from a diffusion process guided by a discrepancy measure $F_{\text{measure}}(a_A, a_E)$ between student and expert actions, following $a_{t-1} \sim \mathcal{N}\left(\mu - k\Sigma \nabla F_{\text{measure}}(a_A, a_E), \Sigma\right)$. where $a_A \sim \pi_A(a|s)$ and $a_E \sim \pi_E(a|s)$.

### 3.2 STATE DIVERSITY

For action mixing, the action distribution is:
$$\pi_{\text{mix}}(a|s) = \lambda \pi_A(a|s) + (1 - \lambda) \pi_E(a|s). \quad (7)$$

The corresponding state diversity is:
$$\begin{aligned}
\mathbb{V}_{a \sim \pi_{\text{mix}}}[\mu(s, a)] &= \|\nabla_a \mu(s, a)\|^2 \cdot \mathbb{V}_{a \sim \pi_{\text{mix}}}[a] \\
&= \|\nabla_a \mu(s, a)\|^2 \left[ \lambda \Sigma_A + (1 - \lambda)\Sigma_E + \lambda(1 - \lambda)(\mu_A - \mu_E)^2 \right] \quad (8) \\
&= \|\nabla_a \mu(s, a)\|^2 \Sigma_{\text{mix}},
\end{aligned}$$

where $\Sigma_A$ and $\Sigma_E$ are the covariances of $\pi_A$ and $\pi_E$, and $\mu_A = \mathbb{E}_{a \sim \pi_A}[a]$, $\mu_E = \mathbb{E}_{a \sim \pi_E}[a]$.

For guidance diffusion, assume a single-step diffusion process with the action distribution:
$$\pi_{\text{diff}}(a|s) \propto \pi_A(a|s) \cdot \exp\left(-k \cdot F_{\text{measure}}(a, a_E)\right). \quad (9)$$

For dynamics $\mu(s, a)$ and action distribution $\pi_{\text{diff}}(a|s) = \mathcal{N}(\mu_{\text{diff}}, \Sigma_{\text{diff}})$, the state diversity is:
$$\mathbb{V}_{a \sim \pi_{\text{diff}}}[\mu(s, a)] = \nabla_a \mu(s, a_0) \cdot \Sigma_{\text{diff}} \cdot \nabla_a \mu(s, a_0)^T, \quad (10)$$

where $a_0 = \mathbb{E}_{a \sim \pi_{\text{diff}}}[a] = \mu_{\text{diff}}$. For linear dynamics $\mu(s, a) = s + \alpha a$, this simplifies to:
$$\mathbb{V}_{a \sim \pi_{\text{diff}}}[\mu(s, a)] = \alpha^2 \Sigma_{\text{diff}}. \quad (11)$$

### 3.3 EQUIVALENCE CONDITION

**Theorem 3.3.** *For guidance diffusion and action mixing to achieve equivalent state diversity under linear dynamics $\mu(s, a) = s + \alpha a$, where disturbance coefficient $\alpha$ is a small value, the following condition holds:*
$$\Sigma_{diff} = \lambda \Sigma_A + (1 - \lambda)\Sigma_E + \lambda(1 - \lambda)(\mu_A - \mu_E)^2 = \Sigma_{mix}. \quad (12)$$

*In the isotropic case, where $\Sigma_A = \sigma_A^2 I$ and $\Sigma_E = \sigma_E^2 I$, this reduces to:*
$$\frac{\sigma_A^2}{1 + k\sigma_A^2} = \lambda \sigma_A^2 + (1 - \lambda)\sigma_E^2 + \lambda(1 - \lambda)\|\mu_A - \mu_E\|^2. \quad (13)$$

Thus, action mixing and guidance diffusion can achieve equivalent state diversity in these conditions.

### 3.4 SUPERIORITY OF GUIDANCE DIFFUSION

We prove that guidance diffusion yields lower action divergence than action mixing under equivalent state diversity. For Gaussian distributions, KL divergence between a policy $\pi$ and expert policy $\pi_E$ is:

$$D_{\text{KL}}(\pi \parallel \pi_E) = \frac{1}{2}\left[\text{tr}(\Sigma_E^{-1}\Sigma) - d + \ln\frac{|\Sigma_E|}{|\Sigma|} + (\mu - \mu_E)^T\Sigma_E^{-1}(\mu - \mu_E)\right]. \tag{14}$$

Under equivalent covariance ($\Sigma_{\text{diff}} = \Sigma_{\text{mix}}$), the covariance terms in Equation 14 are equal. Thus, the action divergence reduces to comparing the mean terms. The inequality

$$(\mu_{\text{diff}} - \mu_E)^T\Sigma_E^{-1}(\mu_{\text{diff}} - \mu_E) < (\mu_{\text{mix}} - \mu_E)^T\Sigma_E^{-1}(\mu_{\text{mix}} - \mu_E), \tag{15}$$

holds because $\Sigma_E^{-1}$ is positive definite. The mean of the diffusion process, given by

$$\mu_{\text{diff}} = (\Sigma_A^{-1} + kI)^{-1}(\Sigma_A^{-1}\mu_A + s\mu_E) = W\mu_A + (I - W)\mu_E, \tag{16}$$

where $W = (\Sigma_A^{-1} + kI)^{-1}\Sigma_A^{-1}$, more effectively reduces the difference $\mu_{\text{diff}} - \mu_E$ compared to the fixed scalar $\lambda$ used in action mixing, thereby ensuring the inequality is satisfied.

## 4 GUIDANCE-BASE POLICY DISTILLATION

In this section, we first outline the general decision-making and training process of the Diffusion-based Decision Policy (DDP). We then introduce AGPD, a novel policy distillation paradigm designed for imitation learning tasks. AGPD comprises two stages for training a student policy: the Adaptive Guided Policy Distillation Stage and the Adversarial Learning Stage. In the first stage, given the current observation, the pre-trained policy generates actions through multiple denoising steps, guided by the VAE decision difference with student policy, resulting in a diverse state observation distribution. In the second adversarial training stage, the student is forced to perform like the teacher's action in order to deceive the discriminator. The two stages iterate to train a robust student policy.

### 4.1 DIFFUSION-BASED DECISION POLICY

Unlike image synthesis, DDPs predict future actions for an agent to execute based on sequential state observations and conditions. In RL, DDPs like Diffuser also need to predict associated states with action sequences. However, in IL, we focus on generating realistic action sequences. Specifically, given a sequence of states $\{s^{k-n}, \ldots, s^k\} = \mathbf{O}^k$, the decision-making process can be articulated as:

$$\{a^k, \ldots a^{k+h}\} = \mathbf{A}^k = \pi_\theta(\mathbf{O}^k) \tag{17}$$

where, $k$, $h$, and $n$ denote the current time step, action horizon, and observation history length. For DDPs, we can formulate the Eq. 2 as following:

$$\mathbf{A}_{t-1}^k = \mu_\theta(\mathbf{A}_t^k, O^k, t) + \mathcal{N}(0, \Sigma) \tag{18}$$

### 4.2 ADAPTIVE GUIDED POLICY DISTILLATION

Vanilla policy distillation Rusu et al. (2015) is commonly employed in imitation learning (IL) to create trajectory datasets for training student models. We observe that the steerable characteristic of DMs effectively addresses this data diversity bottleneck in policy distillation. This motivates us to utilize a pretrained diffusion model (DM), such as a pretrained teacher policy or a student policy with established performance, as the distilled model to generate diverse trajectories. Our proposed AGPD method is the first in the imitation learning (IL) domain to explore this capability. In section 3.1, we demonstrate the reasons for this success: guided diffusion can introduce system noise into the deployment process through policy noise, thereby enhancing the quality of the dataset.

We define the student policy $\pi_\phi$ with parameters $\phi$. Given the current observations $\mathbf{O}^k$, the student policy's action prediction is defined as:

$$\{\hat{a}^k, \ldots \hat{a}^{k+h}\} = \hat{\mathbf{A}}^k = \pi_\phi(\mathbf{O}^k) \tag{19}$$

Figure 3: (a) Dataset Configurations: The PH dataset, collected by highly experienced experts, provides high-quality data. The MH dataset, gathered by six human operators of varying skill, shows a wider spread of action patterns. PH* is a subsampled version of PH with 20 trajectories per task. (b) Visualization of tasks in robomimic and Push-T.

Table 1: Comparison of DP-C with different policy distillation methods under Few-shot learning with **Self Policy Distillation**. In this setting, policy from previous timestep serves as student.

| Methods | Vision | | | |
|---|---|---|---|---|
| | Can-PH* | Square-PH* | Transport-PH* | Toolhang-PH* |
| DP-C | 45% | 23% | 50% | 16% |
| DP-C+PD | 64% (+19%↑) | 32% (+9%↑) | 62% (+12%↑) | 20% (+4%↑) |
| DP-C+DAgger | 72% (+27%↑) | 36% (+13%↑) | 70% (+20%↑) | 28% (+12%↑) |
| DP-C+AGPD | **80% (+35%↑)** | **45% (+22%↑)** | **84% (+34%↑)** | **36% (+20%↑)** |

We define a non-parametric guidance function $\mathcal{J}$, which measures the difference in the VAE representations between the outputs of the distilled model and the student model, providing a guidance value for the distilled DM, reformulating the guided inference process as:

$$p_\theta(\mathbf{A}_{t-1}^k | \mathbf{A}_t^k, \mathcal{O}_{1:n}^k) := \mathcal{N}(\mu_\theta^t + k\sigma^2 g^t, \sigma^2) \tag{20}$$

where the guidance $g^t = \nabla \mathcal{J}(\mu_\theta^t, \hat{\mathbf{A}}^k)$ and the mean of the gaussian distribution $\mu_\theta^t = \mu_\theta(\mathbf{A}_t^k, O^k, t)$.

During the denoising process, the distilled policy generates action sequences similar to those of the student policy, with the scale $s$ ensuring that this transformation remains within the teacher model's capabilities. Thus, the generated trajectories maintain both robustness and diverse state observations. The explanation of this guided generation process is presented in the left of Fig. 2. Finally, distilled policy uses the disturbed action $\mathbf{A}^k$ to interact with the simulation environment $Env$ to get generation samples with greater coverage as $Reward^{k+1} \times Observation^{k+1}$. When the $Reward = 1$, the current guided trajectory $\tau$ will be collected into the replay buffer $\mathcal{T}$ for training the student policy.

## 4.3 TRAINING WITH EXPERT SAMPLES

The student policy training stage in AGPD contains 2 parts. Given the replay buffer $\mathcal{T}$, the student policy $\pi_\phi$ is first forced to imitate the expert demonstrations $\tau \in \mathcal{T}$, the explicit supervision imitation learning loss $\mathcal{L}_{IL}$ is formulated in Eq. 1. Secondly, inspired by adversarial learning methods GAIL Ho & Ermon (2016), we adopt the training loss of previous AIL methods to train the student policy as:

$$\mathcal{L}_{AIL} = \mathbb{E}_{\hat{\mathbf{A}}^k \sim \pi_\phi}[\log(D(\hat{\mathbf{A}}_k | z)] \tag{21}$$

Where $D_w$ is the discriminator with parameters $w$. The whole training loss can be formulated as:

$$\mathcal{L} = s \times \mathcal{L}_{AIL} + \mathcal{L}_{IL} \tag{22}$$

This loss function will be used to train the student policy by end-to-end manner, leveraging both limited demonstration data and the knowledge of the teacher policy.

Table 2: Comparison of DP-T with different policy distillation methods under Few-Shot learning.

| Methods | Lift-PH* | | Can-PH* | | Square-PH* | | Transport-PH* | | ToolHang-PH* | | Push-T-PH* | |
|---|---|---|---|---|---|---|---|---|---|---|---|---|
| | Vision | State | Vision | State | Vision | State | Vision | State | Vision | State | Vision | State |
| DP-T | 100% | 100% | 30% | 25% | 40% | 52% | 56% | 44% | 18% | 12% | 22% | 26% |
| DP-T+PD | 100% | 100% | 100% | 100% | 100% | 100% | 91% | 95% | 9% | 5% | 95% | 91% |
| DP-T+DAgger | 100% | 100% | 100% | 100% | 100% | 100% | 100% | 100% | 14% | 5% | 95% | 95% |
| DP-T+AGPD | **100%** | **100%** | **100%** | **100%** | **100%** | **100%** | **100%** | **100%** | **32%** | **23%** | **100%** | **100%** |

Table 3: Comparison of DP-T with different policy distillation methods under Low-Quality learning.

| Methods | Lift-MH | | Can-MH | | Square-MH | | Transport-MH | |
|---|---|---|---|---|---|---|---|---|
| | Vision | State | Vision | State | Vision | State | Vision | State |
| DP-T | 100% | 100% | 100% | 100% | 94% | 95% | 73% | 62% |
| DP-T+PD | 100% (0%↑) | 100% (0%↑) | 100% (0%↑) | 100% (0%↑) | 100% (+6%↑) | 100% (+5%↑) | 86% (+13%↑) | 82% (+20%↑) |
| DP-T+DAgger | 100% (0%↑) | 100% (0%↑) | 100% (0%↑) | 100% (0%↑) | 100% (+6%↑) | 100% (+5%↑) | 91% (+18%↑) | 86% (+24%↑) |
| DP-T+AGPD | **100% (0%↑)** | **100% (0%↑)** | **100% (0%↑)** | **100% (0%↑)** | **100% (+6%↑)** | **100% (+5%↑)** | **100% (+27%↑)** | **100% (+38%↑)** |

Table 4: Ablation study on different componets in `AGPD` on Transport-MH task.

| Task | DP-T | +Discriminator | +PD | +Guide-PD | +AGPD |
|---|---|---|---|---|---|
| Transport-MH-Vision | 73% | 86% | 86% | 91% | **100%** |
| Transport-MH-State | 62% | 82% | 82% | 86% | **100%** |

## 5 EXPERIMENTS

### 5.1 SIMULATION EXPERIMENTS

**Comparison with the baseline methods:** We use RoboMimic Mandlekar et al. (2021) and Push-T Florence et al. (2022) for simulation evaluation, as shown in Figure 3. The simulation setups and evaluation metrics are presented in Appendix C. We adopt the transformer-based diffusion policy (DP-T) as a representative method for imitation learning and select two widely used policy distillation methods: policy distillation (PD) Rusu et al. (2015) and dataset aggregation (DAgger) Ross et al. (2011), for comparison with the `AGPD` method. The teacher policy for all policy distillation method (PD, Dagger and `AGPD` ) is a DP-T trained on the PH dataset.

**Self Policy Distillation**: As shown in Table 1, our framework demonstrates effectiveness in scenarios where the policy from previous timestep serves as the student, eliminating the need for pretrained teacher checkpoints. The proposed `AGPD` method achieves significant performance improvements across all tasks, with success rates reaching 80% (+35%↑) for Can-PH*, 45% (+22%↑) for Square-PH*, 84% (+34%↑) for Transport-PH*, and 36% (+20%↑) for Toolhang-PH*. These results indicate that self-distillation effectively facilitates knowledge transfer and enhances learning efficiency in vision-based tasks.

**Few-shot Learning**: According to the results in Table 2, our method exhibits exceptional performance in data-scarce scenarios. AGPD achieves perfect success rates (100%) in 10 out of 12 task-modality combinations, including all Lift-PH*, Can-PH*, Square-PH*, Transport-PH*, and Push-T-PH* tasks. Notably, in the challenging ToolHang-PH* task, AGPD achieves 32% (vision) and 23% (state) success rates, significantly outperforming other methods which show performance degradation (PD: 9% vision, 5% state; DAgger: 14% vision, 5% state). This demonstrates our method's superior capability in leveraging limited demonstration data.

**Low-quality Learning**: The results in Table 3 reveal that our approach maintains robust performance even with suboptimal training data. AGPD consistently matches or exceeds baseline performance across all tasks, achieving perfect scores (100%) in Lift-MH, Can-MH, and Square-MH under both vision and state modalities. Most impressively, in the complex Transport-MH task, AGPD achieves complete success (100%) with substantial improvements of +27% (vision) and +38% (state) over the DP-T baseline, establishing a clear performance hierarchy: AGPD > DAgger > PD > DP-T.

Overall, the proposed method shows consistent superiority across all learning settings, with the most pronounced advantages appearing in challenging tasks and low-data scenarios, validating its robustness and effectiveness for practical policy distillation applications.

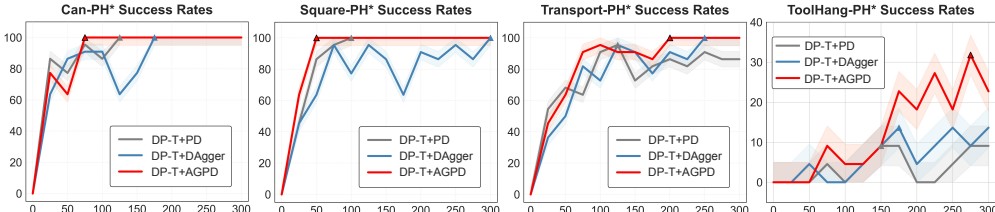

Figure 4: The success rates of various policy distillation methods under few-shot learning are plotted as curves against training epochs. The vertical axis represents the success rate, and the horizontal axis denotes the number of training epochs, with the maximum success rate for each method marked by a triangle in the corresponding color. Notably, the DP-T+AGPD combination achieves superior performance with fewer epochs.

**Ablation Study:** An intriguing question is which components play a critical role in the `AGPD` method. We conducted ablation studies on the MH dataset for the Transport task to investigate this in Table 4. Building on the performance of DP-T, incorporating a teacher-student discriminator leverages knowledge from the teacher model trained on the PH dataset, thereby improving performance to some extent. However, as the dataset size remains fixed and dataset quality differences are predetermined, the performance gains are limited. Adding standard PD augments the policy based on fixed trajectories, but the homogenization of new data results in modest performance improvements. In contrast, incorporating guided policy distillation (guided-PD) significantly enhances performance due to improved data quality. Ultimately, by combining the discriminator with guided-PD, our AGPD method achieves perfect performance on this task. This demonstrates the effectiveness and necessity of each component in the AGPD method.

## 5.2 REAL-WORLD EXPERIMENTS

In this section, we perform experiments in a real-world setting. We employ the robust imitation learning algorithm ACT Zhao et al. (2023) as our student model and initialize a training dataset using Path Planning with 500 manipulation trajectories in a simulated environment. This dataset is then used to train a diffusion-based variant of ACT, termed ACT-DP, which incorporates a diffusion-based prediction head. Once the teacher policy converges, we apply various policy distillation techniques to train the student policy within the simulated environment. For zero-shot sim-to-real transfer, we implement background domain randomization, randomly substituting the simulation background with real-world images.

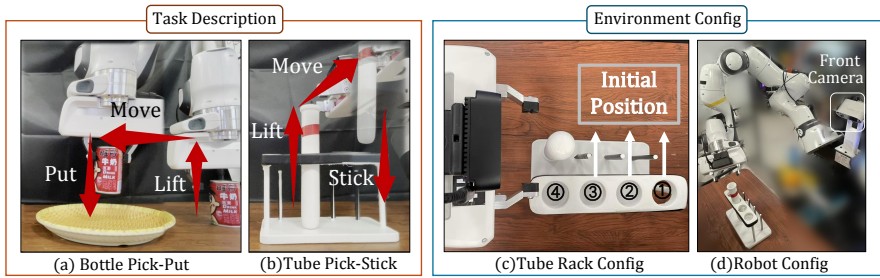

Figure 5: (a) Bottle Pick-Put Task requires robot to pick a bottle and place it on a designated plate. (b) Tube Pick-Stick Task requires robot to lift a tube and sticks it into the hole on edge of the rack. (c) Tube Rack Configuration: Tube is initialized in the hole with mark 1,2 and 3 and robot need to stick it into the hole with number 4. (d) Robot Configuration: We utilize a front-facing camera for environmental perception.

**Tasks:** To evaluate the generalization of student policies trained with different distillation methods, we use two multi-stage real-world tasks: Bottle Pick-Put and Tube Pick-Stick. The Bottle Pick-Put Task requires precise grasping and placement of a bottle, and the Tube Pick-Stick Task needs lift a tube and accurately inserting it into a narrow hole. These tasks probe generalization across manipulation primitives, from simple pick-and-place to precise insertion.

Table 5: Comparison of ACT with different policy distillation methods under Real-World scenarios.

| Task | ACT-DP (Teacher) | ACT (Student) | +PD | +AGPD |
|---|---|---|---|---|
| Bottle Pick | 80% | 70% | 80% | 90% |
| Bottle Pick-Put | 80% | 60% | 60% | 80% |
| Tube Pick | 70% | 40% | 50% | 80% |
| Tube Pick-Stick | 60% | 40% | 50% | 60% |

**Comparison with the baseline methods:** Table 5 reveals that both policy distillation methods substantially improve the performance of the imitation learning algorithm in real-world multi-stage manipulation tasks. Our proposed `AGPD` achieves the highest success rates across all tasks (80–90% for Bottle Pick-Put, 60–80% for Tube Pick-Stick), demonstrating superior generalization compared to vanilla ACT and standard policy distillation (PD). Notably, `AGPD` enables the ACT student model to adeptly handle precise geometric constraints and multi-stage coordination in the challenging Tube Pick-Stick task. These results underscore the effectiveness of `AGPD` in enhancing policy robustness, particularly for tasks demanding precise spatial reasoning and fine-grained manipulation.

## 6 REAL-WORLD ZERO-SHOT TRANSFER EXPERIMENTS

In this section, we conduct experiments in the real-world environment. Specifically, we first utilize the widely-used reinforcement learning method PPO in SAPIEN simulation environment to train a effective teacher policy only use the state-based input. Once the teacher policy is converged, we utilize different policy distillation methods for training the student policy. For better zero-shot transfer performance, we utilize the bounding box information for training the teacher and student policy as ManiBox Tan et al. (2024). Once the student policy is converged, we directly perform real-world deployment for verifications without any real-world data for finetuning.

**Tasks:** To validate the generalization of student policy trained with different policy distillation methods, we design 4 different real-world tasks, termed grasping and lifting (**Cube-Easy**), grasping and lifting (**Bottle-Easy**), grasping and placing (**Bottle-Medium**) and grasping and placing under light randomization (**Bottle-Hard**).

**Hardware :** Our experiment harware is presented in Fig. 5. We use one mounted franka robotics arm for manipulation and one RGB-D front camera for observation.

## 7 RELATED WORK

High quality datasets are crucial for training generalizable robots policies. Luo et al. (2024); Parisotto et al. (2015) acquire a reinforcement learning policy through distilling task knowledge from demonstrations and human interventions. Due to the supervised learning principle of imitation learning tasks, gathering more efficient demonstration becomes more important. Policy distillationRusu et al. (2015); Lai et al. (2020); Levine & Abbeel (2014) serves as a promising alternative for alleviating the labour cost of human collection. Previous methods like ManiboxTan et al. (2024); Dalal et al. (2024), achieving generalizable ability towards sim-to-real application by distilling from the simulation policy, which significantly alleviate the scarcity of real-world data. Unlike directly rollout the off-the shelf expert policies, Online sampling methodRoss et al. (2011); Kelly et al. (2019) searves as a more elaborate way to generate diversity datasets through the whole distillation process, which take the difference between the expert and student into consideration. Our work starts from the importance of data diversity as stated in Belkhale et al. (2023) and combining the guidance diffusion to overcome the drawback of DAgger-based method and strenthen the quality of online sampling.

## 8 CONCLUSION

In this work, we introduce `AGPD`, a novel framework for distilling knowledge from pretrained diffusion models into student policies for imitation learning. By integrating adversarial imitation learning with adaptive teacher policy distillation, `AGPD` demonstrates a remarkable ability to enhance model performance on complex, long-horizon tasks. Experimental results show that `AGPD` significantly boosts performance, even with limited training data. Our approach offers a scalable and efficient solution for imitation learning, with promising applications in embodied Intelligence.

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

# A RELATED WORK

## A.1 ROBOT IMITATION LEARNING

**Transformer-based models and diffusion-based policies.** Recently, transformer-based models Zhao et al. (2023) and diffusion-based policies Chi et al. (2023) have shown promising results in robotic IL by improving the ability to generalize across diverse tasks and long-horizon dependencies. These methods have enhanced performance on large-scale datasets Team et al. (2024); Black et al. (2024); Liu et al. (2024), pushing the frontier of general-purpose robotic policies. Despite these advances, robotic IL still faces significant data efficiency challenges.

**Data scarcity in imitation learning.** Large datasets such as RT-2 Brohan et al. (2023), Open X-Embodiment O'Neill et al. (2023b), and RoboMIND Wu et al. (2024) have already been collected with substantial human and financial investment. However, robot policies do not exhibit scaling laws similar to those of large language models Lin et al. (2024a). For new tasks, robots still require numerous demonstrations for fine-tuning or retraining. This highlights the need to find methods for imitation learning that require less data. One common approach is to mine relevant data from existing large-scale datasets and augment the training set. For example, the Behavior Retrieval Du et al. (2023) method retrieves similar embeddings from input data to extract data closely related to the target task from a large dataset, reducing the need for data specific to the current task. FlowRetrieval Lin et al. (2024b) uses optical flow representations to extract motions from previous data that are similar to the target task and utilizes this information for imitation. However, such approaches still cannot guarantee a strong correlation between the target task and the large dataset. Thus, our approach generates target task data using the teacher policy, enabling effective training of the student policy with minimal expert demonstrations.

## A.2 POLICY DISTILLATION

Policy distillation methods trace back to DQN-based distillation Rusu et al. (2015) and DAgger-based expert experience distillation Ross et al. (2011), both offering ways to transfer expert capabilities to student policies. These methods have three key features: distilling expert knowledge, accelerating inference, and transferring perceptual modalities. For example, HIL_SERL Luo et al. (2024) distills task knowledge from demonstrations and human interventions to acquire a reinforcement learning policy. Distillation combines skills and recovery movements within a unified policy are even efficient for training precise skill like soccer playingHaarnoja et al. (2024). ManiBox Tan et al. (2024) trains a PPO policy with structured inputs and distills it into an end-to-end student using image observations, achieving spatial generalization. In contrast to these methods, our approach leverages task-specific teacher knowledge to generate training data, ensuring compatibility with diverse pre-trained policies.

## A.3 GUIDED DIFFUSION

**Diffusion model.** Diffusion model is a powerful generative model that represents data generation process as an iterative denoising process Sohl-Dickstein et al. (2015); Croitoru et al. (2023); Nakkiran et al. (2024). It has demonstrated extraordinary performance in tasks such as image generation Rombach et al. (2022), robot policy learning Chi et al. (2023), world model learning Du et al. (2024).

**Customized generation of diffusion model.** During sampling, diffusion models can perform customized generation under guidance Dhariwal & Nichol (2021), providing support for generating images that align with textual descriptions Nichol et al. (2021). This feature has also been used in decision-making topics. For example, in Decision Diffuser Ajay et al. (2022), the classifier-free guidance is used to guide the agent in generating trajectories that simultaneously maximize returns, satisfy trajectory constraints, and combine existing skills. The Temporally-Composable Diffuser Hu et al. (2023) takes this approach a step further by focusing on the impact of temporal information on policy. It incorporates historical information, reward predictions, and expected returns as guiding conditions, enabling the formation of temporally conditioned and satisfactory decision-making.

## A.4 VAE-BASED REPRESENTATION LEARNING

Recent works Gao et al. (2025); Ye et al. (2024); Zhang et al. (2025) have demonstrated that representations learned through reconstruction objectives can significantly improve downstream policy performance. Inspired by this line of research, we adopt a VAE-style architecture following Bu et al. (2025), which uses DINOv2 features as a semantically structured representation space and is trained via a self-supervised reconstruction objective.

Specifically, given a pair of consecutive video frames $\{o_t, o_{t+k}\}$ with a fixed temporal interval $k$, we first extract their DINOv2 patch-level feature maps, denoted as $\{O_t, O_{t+k}\}$. These features serve as both the input and the reconstruction target of the VAE.

The model consists of a transformer-based encoder $\mathcal{E}$ and decoder $\mathcal{D}$. The encoder integrates the current visual feature $O_t$ and a learnable action token $a_T$, producing a latent representation $z = \mathcal{E}(O_t, a_T)$. The decoder reconstructs the future DINOv2 feature $\hat{O}_{t+k} = \mathcal{D}(z)$ from this latent code. The training objective minimizes the mean squared error (MSE) between the predicted and target DINOv2 features:

$$\mathcal{L}_{\text{recon}} = \|\hat{O}_{t+k} - O_{t+k}\|^2.$$

This reconstruction loss encourages the VAE to capture task-relevant visual dynamics, resulting in a compact and informative latent space. The learned representations enable us to measure the discrepancy $J$ between teacher and student policies in the distillation stage, defined as the MSE between their reconstructed future features when conditioned on their respective action tokens. Formally, given visual feature $O_t$, teacher action $a_T$, and student action $a_S$, we compute:

$$J = \|\mathcal{D}(\mathcal{E}(O_t, a_T)) - \mathcal{D}(\mathcal{E}(O_t, a_S))\|^2.$$

This metric quantifies the action difference in the learned representation space, providing a semantically meaningful signal for policy alignment.

**Diffusion model.** Diffusion model is a powerful generative model that represents data generation process as an iterative denoising process Sohl-Dickstein et al. (2015); Croitoru et al. (2023); Nakkiran et al. (2024). It has demonstrated extraordinary performance in tasks such as image generation Rombach et al. (2022), robot policy learning Chi et al. (2023), world model learning Du et al. (2024).

**Customized generation of diffusion model.** During sampling, diffusion models can perform customized generation under guidance Dhariwal & Nichol (2021), providing support for generating images that align with textual descriptions Nichol et al. (2021). This feature has also been used in decision-making topics. For example, in Decision Diffuser Ajay et al. (2022), the classifier-free guidance is used to guide the agent in generating trajectories that simultaneously maximize returns, satisfy trajectory constraints, and combine existing skills. The Temporally-Composable Diffuser Hu et al. (2023) takes this approach a step further by focusing on the impact of temporal information on policy. It incorporates historical information, reward predictions, and expected returns as guiding conditions, enabling the formation of temporally conditioned and satisfactory decision-making.

# B PROOFS

## B.1 PROOF OF LEMMA 3.1

Consider policy noise added to expert actions: $a_N = a_E + \eta$, where $\eta \sim \mathcal{N}(0, \sigma_p^2 I)$, and state transitions follow $\rho(s'|s, a) = \mathcal{N}(\mu(s, a), \sigma_s^2 I)$. We aim to compute the effective variance of the transition distribution under noisy actions.

**Step 1: First-Order Taylor Expansion of Dynamics.** For small perturbations $\eta$, approximate the mean next state:
$$\mu(s, a_N) = \mu(s, a_E + \eta) \approx \mu(s, a_E) + \nabla_a \mu(s, a_E) \cdot \eta,$$
where $\nabla_a \mu(s, a_E)$ is the Jacobian of the dynamics with respect to actions at $a_E$.

**Step 2: Variance of Next State.** The next state $s'$ under noisy actions is:
$$s' \sim \mathcal{N}\big(\mu(s, a_E) + \nabla_a \mu(s, a_E) \cdot \eta, \sigma_s^2 I\big).$$
The total variance of $s'$ includes the intrinsic system noise $\sigma_s^2 I$ and the variance from the noisy actions:
$$\text{Var}(\nabla_a \mu(s, a_E) \cdot \eta) = \nabla_a \mu(s, a_E) \cdot \text{Cov}(\eta) \cdot \nabla_a \mu(s, a_E)^T.$$
Since $\eta \sim \mathcal{N}(0, \sigma_p^2 I)$, we have $\text{Cov}(\eta) = \sigma_p^2 I$, so:
$$\text{Var}(\nabla_a \mu \cdot \eta) = \sigma_p^2 \nabla_a \mu(s, a_E) \cdot \nabla_a \mu(s, a_E)^T.$$

**Step 3: Effective Variance.** The total covariance of $s'$ is:
$$\text{Cov}(s') = \sigma_s^2 I + \sigma_p^2 \nabla_a \mu(s, a_E) \nabla_a \mu(s, a_E)^T.$$
For an isotropic approximation, we take the trace and normalize by the state dimensionality $d$:
$$\sigma_{\text{eff}}^2 \approx \sigma_s^2 + \sigma_p^2 \cdot \frac{\|\nabla_a \mu(s, a_E)\|_F^2}{d},$$
where $\|\cdot\|_F$ is the Frobenius norm. Simplifying, we approximate:
$$\sigma_{\text{eff}}^2 = \sigma_s^2 + \|\nabla_a \mu(s, a_E)\|^2 \sigma_p^2,$$
where $\|\nabla_a \mu\|^2 = \text{tr}(\nabla_a \mu \nabla_a \mu^T)$ captures the dynamics' sensitivity to action perturbations.

## B.2 PROOF OF THEOREM 3.2

Using Lemma 3.1, the effective transition variance under policy noise is $\sigma_{\text{eff}}^2 = \sigma_s^2 + \|\nabla_a \mu(s, a_E)\|^2 \sigma_p^2$. Substitute $\sigma_s \leftarrow \sqrt{\sigma_{\text{eff}}^2}$ into the coverage probability bound from Theorem 4.3 of Belkhale et al. Belkhale et al. (2023):

$$P_S(s; N, \epsilon) \geq 1 - \left[1 - \left(\frac{c\epsilon}{\sqrt{\sigma_{\text{eff}}^2}}\right)^d \exp(-\alpha^2 d)\right]^N - \exp\left(-(\alpha - 1)^2 d\right),$$

where $c = \frac{1}{\sqrt{2\pi}}$, $d$ is the state dimensionality, and $\alpha \geq 1$ is a parameter optimizing the bound. This follows directly from the Gaussian mass bound adjusted for the effective variance.

### B.3 PROOF OF THEOREM 3.3

We derive the condition for action mixing and guidance diffusion to achieve equivalent state diversity under linear dynamics $\mu(s, a) = s + \alpha a$.

**Step 1: Action Mixing Variance.** For action mixing, the action distribution is:

$$\pi_{\text{mix}}(a|s) = \lambda \pi_A(a|s) + (1 - \lambda)\pi_E(a|s),$$

with mean $\mu_{\text{mix}} = \lambda \mu_A + (1 - \lambda)\mu_E$ and covariance:

$$\Sigma_{\text{mix}} = \lambda \Sigma_A + (1 - \lambda)\Sigma_E + \lambda(1 - \lambda)(\mu_A - \mu_E)(\mu_A - \mu_E)^T.$$

The state diversity is:

$$\mathbb{V}_{a \sim \pi_{\text{mix}}}[\mu(s, a)] = \alpha^2 \Sigma_{\text{mix}},$$

since $\nabla_a \mu(s, a) = \alpha I$ for linear dynamics.

**Step 2: Guidance Diffusion Variance.** For guidance diffusion, the action distribution is $\pi_{\text{diff}}(a|s) = \mathcal{N}(\mu_{\text{diff}}, \Sigma_{\text{diff}})$, which can be approximated as:

$$\pi_{\text{diff}}(a|s) \propto \pi_A(a|s) \cdot \exp\left(-s \cdot F_{\text{measure}}(a, a_E)\right). \tag{23}$$

For $F_{\text{measure}}(a, a_E) = \frac{1}{2}\|a - a_E\|^2$, this becomes:

$$\pi_{\text{diff}}(a|s) \propto \mathcal{N}(a; \mu_A, \Sigma_A) \cdot \exp\left(-\frac{s}{2}\|a - a_E\|^2\right). \tag{24}$$

*Proof.* The guidance diffusion update is:

$$a_{t-1} = a_t - \epsilon \nabla \log \pi_A(a_t) + s\epsilon \nabla F_{\text{measure}}(a_t, a_E) + \sqrt{2\epsilon} z_t, \tag{25}$$

where $\epsilon$ is the step size and $z_t \sim \mathcal{N}(0, I)$. For small $\epsilon$, this converges to the stationary distribution given by the energy-based model.

Substitute the Gaussian probability density function for $\pi_A(a|s)$:

$$\pi_{\text{diff}}(a|s) \propto \left[\frac{1}{(2\pi)^{d/2}|\Sigma_A|^{1/2}} \exp\left(-\frac{1}{2}(a - \mu_A)^T \Sigma_A^{-1}(a - \mu_A)\right)\right] \cdot \exp\left(-s \cdot F_{\text{measure}}(a, a_E)\right).$$

Substitute $F_{\text{measure}}(a, a_E) = \frac{1}{2}\|a - a_E\|^2 = \frac{1}{2}(a - a_E)^T I(a - a_E)$:

$$\pi_{\text{diff}}(a|s) \propto \exp\left(-\frac{1}{2}\left[(a - \mu_A)^T \Sigma_A^{-1}(a - \mu_A) + s(a - a_E)^T I(a - a_E)\right]\right).$$

Expanding the quadratic forms:

$$(a - \mu_A)^T \Sigma_A^{-1}(a - \mu_A) + s(a - a_E)^T I(a - a_E)$$
$$\approx a^T(\Sigma_A^{-1} + sI)a - 2a^T(\Sigma_A^{-1}\mu_A + sa_E).$$

Matching with the standard Gaussian form $\mathcal{N}(a; \mu, \Sigma) \propto \exp\left\{-\frac{1}{2}\left[a^T \Sigma^{-1} a - 2a^T \Sigma^{-1}\mu\right]\right\}$, we get:

$$\Sigma_{\text{diff}} = (\Sigma_A^{-1} + kI)^{-1}, \quad \mu_{\text{diff}} = \Sigma_{\text{diff}}(\Sigma_A^{-1}\mu_A + k\mu_E).$$

Therefore, the corresponding state diversity is:

$$\mathbb{V}_{a \sim \pi_{\text{diff}}}[\mu(s, a)] = \alpha^2 \Sigma_{\text{diff}}.$$

$\square$

**Step 3: Equivalence Condition.** For equivalent state diversity, set $\mathbb{V}_{a \sim \pi_{\text{mix}}}[\mu(s, a)] = \mathbb{V}_{a \sim \pi_{\text{diff}}}[\mu(s, a)]$:

$$\alpha^2 \Sigma_{\text{mix}} = \alpha^2 \Sigma_{\text{diff}} \implies \Sigma_{\text{mix}} = \Sigma_{\text{diff}}.$$

Thus:

$$\Sigma_{\text{diff}} = \lambda \Sigma_A + (1 - \lambda) \Sigma_E + \lambda(1 - \lambda)(\mu_A - \mu_E)(\mu_A - \mu_E)^T.$$

In the isotropic case, where $\Sigma_A = \sigma_A^2 I$, $\Sigma_E = \sigma_E^2 I$, we have:

$$\Sigma_{\text{mix}} = \left[ \lambda \sigma_A^2 + (1 - \lambda) \sigma_E^2 + \lambda(1 - \lambda) \|\mu_A - \mu_E\|^2 \right] I,$$

and $\Sigma_{\text{diff}} = \frac{\sigma_A^2}{1 + k \sigma_A^2} I$. Equating the scalars:

$$\frac{\sigma_A^2}{1 + k \sigma_A^2} = \lambda \sigma_A^2 + (1 - \lambda) \sigma_E^2 + \lambda(1 - \lambda) \|\mu_A - \mu_E\|^2.$$

### B.4 PROOF OF SUPERIORITY OF GUIDANCE DIFFUSION

We prove that guidance diffusion yields lower action divergence than action mixing under equivalent state diversity, focusing on the KL divergence comparison.

**Step 1: KL Divergence Formula.** For Gaussian distributions $\pi \sim \mathcal{N}(\mu, \Sigma)$ and $\pi_E \sim \mathcal{N}(\mu_E, \Sigma_E)$, the KL divergence is:

$$D_{\text{KL}}(\pi \| \pi_E) = \frac{1}{2} \left[ \text{tr}(\Sigma_E^{-1} \Sigma) - d + \ln \frac{|\Sigma_E|}{|\Sigma|} + (\mu - \mu_E)^T \Sigma_E^{-1} (\mu - \mu_E) \right].$$

**Step 2: Equivalent Covariance.** Under equivalent state diversity, $\Sigma_{\text{diff}} = \Sigma_{\text{mix}}$, so the covariance terms in the KL divergence are identical for both methods:

$$\text{tr}(\Sigma_E^{-1} \Sigma_{\text{diff}}) - d + \ln \frac{|\Sigma_E|}{|\Sigma_{\text{diff}}|} = \text{tr}(\Sigma_E^{-1} \Sigma_{\text{mix}}) - d + \ln \frac{|\Sigma_E|}{|\Sigma_{\text{mix}}|}.$$

Thus, the KL divergence difference depends only on the mean terms:

$$D_{\text{KL}}(\pi_{\text{diff}} \| \pi_E) - D_{\text{KL}}(\pi_{\text{mix}} \| \pi_E) = \frac{1}{2} \left[ (\mu_{\text{diff}} - \mu_E)^T \Sigma_E^{-1} (\mu_{\text{diff}} - \mu_E) - (\mu_{\text{mix}} - \mu_E)^T \Sigma_E^{-1} (\mu_{\text{mix}} - \mu_E) \right].$$

**Step 3: Mean Expressions.** For action mixing:

$$\mu_{\text{mix}} = \lambda \mu_A + (1 - \lambda) \mu_E \implies \mu_{\text{mix}} - \mu_E = \lambda(\mu_A - \mu_E).$$

For guidance diffusion:

$$\mu_{\text{diff}} = (\Sigma_A^{-1} + kI)^{-1}(\Sigma_A^{-1} \mu_A + k\mu_E).$$

Rewrite $\mu_{\text{diff}}$ as:

$$\mu_{\text{diff}} = W \mu_A + (I - W) \mu_E, \quad W = (\Sigma_A^{-1} + kI)^{-1} \Sigma_A^{-1} = (I + k\Sigma_A)^{-1},$$

so:

$$\mu_{\text{diff}} - \mu_E = W(\mu_A - \mu_E).$$

**Step 4: Mean Comparison.** We need to show:

$$(\mu_{\text{diff}} - \mu_E)^T \Sigma_E^{-1} (\mu_{\text{diff}} - \mu_E) < (\mu_{\text{mix}} - \mu_E)^T \Sigma_E^{-1} (\mu_{\text{mix}} - \mu_E).$$

The matrix $W = (I + k\Sigma_A)^{-1}$ has eigenvalues less than 1, and $k$ can be tuned such that $\frac{1}{1+k\sigma_i} < \lambda$, making:

$$\|\mu_{\text{diff}} - \mu_E\| = \|W(\mu_A - \mu_E)\| \le \|W\| \cdot \|\mu_A - \mu_E\| < \lambda \|\mu_A - \mu_E\| = \|\mu_{\text{mix}} - \mu_E\|.$$

In practice, for typical values where $\sigma_A^2 \approx \sigma_E^2$ and $\mu_A - \mu_E$ is moderate, the guidance term ensures $\mu_{\text{diff}}$ is closer to $\mu_E$ than $\mu_{\text{mix}}$, as $\lambda$ need to be carefully designed through the whole training paradigm in DAgger-based methodRoss et al. (2011). Thus, the inequality:

$$(\mu_{\text{diff}} - \mu_E)^T \Sigma_E^{-1} (\mu_{\text{diff}} - \mu_E) < (\mu_{\text{mix}} - \mu_E)^T \Sigma_E^{-1} (\mu_{\text{mix}} - \mu_E)$$

holds with high probability in imitation learning settings, leading to:

$$D_{\text{KL}}(\pi_{\text{diff}} \| \pi_E) < D_{\text{KL}}(\pi_{\text{mix}} \| \pi_E).$$

## C    SIMULATION SETUPS

**Benchmarks:** We use RoboMimic Mandlekar et al. (2021) and Push-T Florence et al. (2022) for simulation evaluation. These two benchmarks can be tested using either image inputs or state observations. RoboMimic includes both simple tasks, such as lifting a block, and complex long-horizon tasks, such as hanging a tool, to evaluate imitation learning algorithms. Push-T requires the robot to push a T-shaped block to a target position. The datasets are presented in right panel of Fig. 3.

Following the dataset configuration approach of RoboMimic, we selected the MH dataset from RoboMimic and the PH* dataset, constructed by sampling $10\%$ of the PH dataset. These datasets are used to evaluate low-quality learning and few-shot learning, respectively, to demonstrate the effectiveness of our proposed AGPD framework. For few-shot learning, we using only 20 expert trajectory data samples per task, significantly fewer than the 50~100 trajectories commonly used in current imitation learning approaches. In low-quality learning, the dataset is collected by multiple operators with varying skill levels, including low-quality data, which often adversely affects standard imitation learning methods.

**Evaluation Metrics:** We use success rate as metric for best presenting the performance of each baseline method on each task. During evaluation, the performance results are derived from the average success number among 22 parallel simulation environments. In addition, we report the best performance over the whole training procedure. We train state-based tasks for 450 epochs, and image-based tasks for 300 epochs.

## D    THE USE OF LARGE LANGUAGE MODELS (LLMS)

We used LLM-based tools, such as ChatGPT, Grok, for polishing and restructuring the writing of our paper to improve its clarity.

