# OpenReview forum: "AGPD: Adaptive Guidance policy distillation for Imitation Learning"
_ICLR.cc/2026/Conference — Submitted to ICLR 2026_

### Official Review · Reviewer_VLXC · 2025-10-29

**Soundness:** 3
**Presentation:** 1
**Contribution:** 3
**Rating:** 4
**Confidence:** 4

**Summary:**

This paper introduces AGPD (Adaptive Guidance Policy Distillation), a framework addressing data scarcity in imitation learning through policy distillation. The core idea is to use a pretrained diffusion model as a teacher that generates diverse trajectories guided by the discrepancy between student and teacher policies in VAE representation space. The approach combines guided diffusion for data generation with adversarial learning to encourage the student to mimic teacher behavior.
The paper makes several contributions: (1) an adaptive guidance mechanism based on VAE representation differences, (2) theoretical analysis showing that guidance diffusion achieves smaller action divergence than action mixing under equivalent state diversity (Theorem 3.4), (3) a discriminator-based adversarial learning component, and (4) experimental validation on RoboMimic, Push-T, and real robot tasks demonstrating effectiveness in few-shot and low-quality data scenarios. The work builds upon Belkhale et al. (2023)'s theory connecting system noise to state diversity, extending this insight to policy distillation.

**Strengths:**

The paper addresses an important challenge in imitation learning where data scarcity causes compounding errors. The motivation is well-grounded in recent theory (Belkhale et al. 2023) and correctly identifies key limitations of existing methods - deterministic teacher behavior in traditional PD and low-quality mixing in DAgger.

The method design shows thoughtful engineering with a clear two-stage framework combining guided generation for exploration with adversarial learning for stability. The self-distillation capability is particularly valuable, enabling iterative policy improvement without pretrained teachers.

The simulation experiments are comprehensive, evaluating few-shot learning (20 trajectories per task), mixed-quality data, both vision and state modalities, and tasks of varying difficulty. The results demonstrate substantial improvements with AGPD consistently outperforming baselines, and Figure 4 shows faster convergence in addition to better final performance.

**Weaknesses:**

The theoretical analysis relies on a linear dynamics assumption (μ(s, a) = s + αa) that is clearly violated by the contact-rich manipulation tasks evaluated. While this assumption is stated explicitly (line 208-209), the paper never discusses its validity or how conclusions might extend to realistic nonlinear settings. Most critically, there is no experimental validation of the theoretical predictions - no measurements of actual KL divergences to verify Theorem 3.4, no computation of ||∇_aμ||² to validate Lemma 3.1. This gap between theory and practice suggests the analysis provides post-hoc intuition rather than principled design guidance.

Real-world validation is insufficient for a method with this level of complexity. Only two simple tabletop tasks are evaluated, and essential technical details are missing or unclear. The paper states the teacher is trained in simulation (lines 408-410) but never clarifies where guided generation occurs or how the sim-trained teacher operates on the real robot. AGPD combines multiple components (guided diffusion, discriminator, VAE representation) that each could introduce sim-to-real gaps, yet there are no real-world ablations to determine which components actually contribute. The complex pipeline makes it particularly important to verify effectiveness in realistic settings beyond tabletop manipulation.

The experimental setup raises fairness concerns. AGPD filters trajectories to keep only successful ones (Reward=1, line 307), but it's unclear if baselines use similar filtering. If AGPD generates many trajectories and keeps only successful ones while baselines use all generated data, this conflates the effects of guided generation with aggressive data curation. The paper doesn't report how many trajectories are generated versus retained, making performance comparisons difficult to interpret.

The ablation study (Table 4) covers only one task, insufficient to establish that components are necessary across diverse scenarios. Given the wide performance range - some tasks reach 100% while ToolHang-PH* achieves only 32%/23% - understanding which components help in different difficulty regimes and why the method fails on hard tasks would significantly strengthen the evaluation.

Beyond these substantive concerns, the organizational confusion between Section 5.2 ("Real-World Experiments") and Section 6 ("Real-World Zero-Shot Transfer Experiments") raises questions about the overall rigor of the paper's preparation. These sections appear to describe overlapping experiments with inconsistent framing and incomplete technical details, suggesting the manuscript may benefit from more careful revision and verification throughout. When key experimental sections contain such ambiguities, it raises concern about whether other parts of the paper might similarly contain unclear or inconsistent descriptions that could affect reproducibility and interpretation.

**Questions:**

On theoretical assumptions and validation: The linear dynamics assumption is clearly unrealistic for contact-rich manipulation. Can you discuss when this provides a reasonable approximation and whether conclusions extend qualitatively to nonlinear settings?

On real-world experimental setup: The relationship between Section 5.2 and Section 6 is confusing - are these the same experiments or different setups? More critically, where does guided generation occur (simulation or real robot)? If in simulation, how do you ensure generated data transfers to real deployment? If on the real robot, what safety mechanisms prevent dangerous exploratory actions from the diffusion policy? Can you provide a complete pipeline diagram showing the data flow for real-world experiments, including where each component operates?

On experimental fairness: AGPD filters trajectories to keep only successful ones (Reward=1). Do baseline methods use similar filtering? How many trajectories are generated during rollouts, and what fraction pass the filter? This is essential to understand whether performance gains come from the guidance mechanism or from more selective data curation.

On ablations and failure analysis: Table 4's ablation covers only Transport-MH. Can you provide ablations for other tasks, particularly challenging ones like ToolHang-PH* with 32%/23% success? What are the primary failure modes in these difficult tasks, and do the discriminator and guided generation components help?

---

### Official Review · Reviewer_6yCc · 2025-10-31

**Soundness:** 3
**Presentation:** 2
**Contribution:** 2
**Rating:** 2
**Confidence:** 4

**Summary:**

This paper introduces the AGPD framework, which aims to enhance imitation learning in two challenging scenarios: few-shot learning and learning from low-quality demonstrations. The core of AGPD is a novel distillation process that transfers knowledge from a pre-trained diffusion model into a student policy. Unlike conventional approaches, the diffusion-based teacher in AGPD explicitly models the divergence between the teacher and student policies to generate informative samples. Furthermore, AGPD incorporates a discriminator to assess policy quality, guiding the student to emulate teacher behaviors that most effectively facilitate learning from mixed-quality datasets.

**Strengths:**

- Relevance and motivation: Tackling few-shot imitation learning and robustness to low-quality demonstrations are timely and practically important problems.
- The description of the algorithmic approach is detailed, and the experiments are comprehensive. The method provides a scalable and relatively robust estimator of task performance, with real-robot experiments further validating its effectiveness.

**Weaknesses:**

- **Overall preparedness appears insufficient:**  The paper’s core contributions and intended application needs are not clearly articulated. The authors should explicitly position their method relative to (a) diffusion policy distillation or consistency-model–based policies, (b) demonstration-augmented RL, and (c) data or trajectory generation approaches. The application scenario is unclear—e.g., how the teacher policy’s training data are obtained and whether simply finetuning the teacher policy is feasible or preferable.
- **Architectural clarity is lacking:** The network structures of the teacher and student policies are not consistently or sufficiently described. Figure 2 reportedly depicts a diffusion model as a UNet, while the experiments describe using a Transformer-based diffusion policy; the authors should clarify whether they employ a Transformer-based UNet and detail the encoder/decoder design, conditioning signals, and action parameterization.
- **Writing quality and minor errors:** The paper contains numerous phrasing issues and minor errors that hinder readability and credibility.

**Questions:**

- Clarify the problem statement and contributions: The paper should clearly define the target application scenarios, tasks, and constraints. The core contributions should be explicitly motivated by and aligned with these practical requirements.

- Strengthen positioning against relevant baselines: The related work and experiments should systematically compare and position the proposed method against three key categories of approaches: (1) Diffusion Policy distillation/consistency model-based policies, (2) demonstration-augmented RL methods, and (3) data/trajectory generation techniques. Include strong and representative baselines from these categories, ensuring all are tuned fairly.

- Provide a more thorough and transparent Sim2Real evaluation: The simulation environment design and assumptions should be described in sufficient detail to ensure reproducibility. For real-robot validation, include a video demonstration of the hardware experiments to better illustrate practical performance and deployment.

- Improve writing consistency and clarity: Standardize terminology throughout the manuscript, correct inconsistencies and minor errors, streamline the narrative to eliminate redundancy, and enhance overall readability.
Minor error:
Figure 3: PH is a subsampled... should be Ph* is ...
Table 1: What is DP-C?
374 line in Page 7: punctuation error

---

### Official Review · Reviewer_8sk1 · 2025-10-31

**Soundness:** 3
**Presentation:** 3
**Contribution:** 2
**Rating:** 4
**Confidence:** 3

**Summary:**

This paper provide a solution for the imitation learning especially when expert demonstration is lack. Specifically, the paper first use diffusion model to generate sythentic trajectories, serving as teacher policy. Then, with synthetic trajectories and the expert trajectories, they learn a student policy. Also, they propose to use gail style imitation learning strategy as well as MSE to learn the student policy. Theoretical results show that the diffusion model can improve the diversity of the training trajectories thus improve the student policy's performance.

**Strengths:**

The theoretical analysis is well-connected with the algorithm. The motivation is clear, aiming to generate more diverse teacher data, and the presentation is effective. Additionally, the guided diffusion appears novel, aiming to generate data more similar to student policy to enhance training. The real-world experiment is good.

**Weaknesses:**

1. The method seems to be complicated and maybe hard to tune, with several components like guided DM and discriminator training, which are known for being hard to tune, especially for the discriminator. There are also two action injection parts, why do you need two action injection? How do you balance that, and how does each component contribute to the results?

2. The computation cost might be high.

3. What do guided diffusion generate the trajectories look like? Are they always successful or have a high success rate? How close is it to the student trajectories?

**Questions:**

See weakness.

---

### Official Review · Reviewer_oabY · 2025-11-01

**Soundness:** 2
**Presentation:** 3
**Contribution:** 3
**Rating:** 6
**Confidence:** 3

**Summary:**

The paper proposes a two‑stage distillation framework. Stage 1 uses a diffusion‑policy teacher guided toward the student via a VAE‑based discrepancy to generate trajectories. Stage 2 trains a discriminator to separate teacher vs student actions and adds an adversarial imitation term to the student loss. Theory argues that guidance diffusion achieves the same state diversity as action mixing while yielding lower action divergence to the expert under linear‑Gaussian assumptions. Simulation on RoboMimic and Push‑T and two real‑robot tasks report good performance, especially in few‑shot and mixed‑quality settings.

**Strengths:**

1. Clear motivation to use guidance diffusion: The paper compare the difference between using mixture of action and guidance diffusion with strong and formal statement.

2. Exceptional Empirical Validation: The empirical results are robust and demonstrate a good performance across all tested scenarios, especially in the most challenging few-shot and low-quality data settings in high-precision robotic manipulation tasks

3. Ablation Study: The ablation study in Table 4 effectively demonstrates the contribution of each component, providing evidence that the full AGPD pipeline is necessary for the best performance.

**Weaknesses:**

1. No implemntation detail: The paper gives no VAE architecture, dataset, loss, or training protocol, and provides no appendix details.

2. Undefined notaion: At line 295, the author only states the diffusion guidance function $J$ is measured from the latent space in the VAE, without specific defintion.

3. Notational ambiguity: The paper use $\Alpah$ as notion of action in Eqn 4 and 5 while using $a$ in the rest of the paper. Please change the action notion in preliminary and Eqn 5.

**Questions:**

Same as weakness.

additional questions

1. Can you provide a precise definition of the "VAE representation divergence $J$"? What is the architecture of the VAE, what is its input/output, and how is it trained?

2. can you correct typos in line 249, caption of Fig3 (I believe a PH* is subsampled) and line 375.

---

### Meta-Review · Area_Chair_VVXc · 2025-12-26

**Summary:**

I have read the paper twice, and still cannot understand the very basics of it: what is the problem statement? What are the algorithms? What are the inputs and outputs to the method?
The reviews are equally confusing, with each summary being very very different.

While it may be that this paper contains an interesting contribution, its current presentation is way below the bar for ICLR. I urge the authors to clearly write their problem definition, algorithm (pseudocode would be great, also architectures), and assumptions.

**Reviewer Concerns:**

see above

**Reviewer Scores:**

6,4,2,4

---

### Decision · Program_Chairs · 2026-01-26

Reject